# Neuroprotective Panel of Olive Polyphenols: Mechanisms of Action, Anti-Demyelination, and Anti-Stroke Properties

**DOI:** 10.3390/nu14214533

**Published:** 2022-10-28

**Authors:** Tanja Grubić Kezele, Božena Ćurko-Cofek

**Affiliations:** 1Department of Clinical Microbiology, Clinical Hospital Center Rijeka, Krešimirova 42, 51000 Rijeka, Croatia; 2Department of Physiology, Immunology and Pathophysiology, Faculty of Medicine, University of Rijeka, Braće Branchetta 20, 51000 Rijeka, Croatia

**Keywords:** cerebrovascular diseases, demyelination, extra virgin olive oil, multiple sclerosis, neuroprotection, olive leaves, olive leaf extract, olive oil, olive polyphenols, stroke

## Abstract

Neurological diseases such as stroke and multiple sclerosis are associated with high morbidity and mortality, long-term disability, and social and economic burden. Therefore, they represent a major challenge for medical treatment. Numerous evidences support the beneficial effects of polyphenols from olive trees, which can alleviate or even prevent demyelination, neurodegeneration, cerebrovascular diseases, and stroke. Polyphenols from olive oils, especially extra virgin olive oil, olive leaves, olive leaf extract, and from other olive tree derivatives, alleviate inflammation and oxidative stress, two major factors in demyelination. In addition, they reduce the risk of stroke due to their multiple anti-stroke effects, such as anti-atherosclerotic, antihypertensive, antioxidant, anti-inflammatory, hypocholesterolemic, hypoglycemic, and anti-thrombotic effects. In addition, olive polyphenols have beneficial effects on the plasma lipid profiles and insulin sensitivity in obese individuals. This review provides an updated version of the beneficial properties and mechanisms of action of olive polyphenols against demyelination in the prevention/mitigation of multiple sclerosis, the most common non-traumatic neurological cause of impairment in younger adults, and against cerebral insult with increasing incidence, that has already reached epidemic proportions.

## 1. Introduction

The olive tree (*Olea europaea* L.) is a valuable source of polyphenols, including all olive derivatives. Of all olive oil products, extra virgin olive oil (EVOO) contains the highest amount of total phenols (200–800 mg/kg) [1]. Olive leaves and olive leaf extract have even higher concentrations of total phenols than olive fruit and olive oil [2,3,4]. Moreover, it is known that the olive leaf is a traditional raw material used in phytotherapy [5]. Polyphenols from virgin olive oil or EVOO, olive leaves, and leaf extract, and other polyphenols derived from the olive tree, have been widely identified in experimental, clinical, and epidemiological research as compounds that can exert many different beneficial effects in the body, especially with long-term consumption [6]. The most studied polyphenols include oleuropein, oleocanthal, oleacein, hydroxytyrosol, and tyrosol.

Polyphenols are organic compounds with different chemical structures and biological activities. Based on their chemical structure, we can distinguish about 8000 polyphenols. They have an aromatic ring and at least one hydroxyl group as a key structure [7]. Three groups of polyphenols are most often held responsible for biological effects: secoiridoid derivatives, phenolic alcohols, and phenolic acids [8]. All of these major phenolic subclasses are present in olive derivatives, as are flavonoids and lignans [9]. Quantitatively, secoiridoids are the most abundant [10]. The main secoiridoids are oleuropein and oleocanthal, while the main phenolic alcohols are hydroxytyrosol and tyrosol [11].

The beneficial effects of polyphenols on human health depend on their absorption, metabolism, and bioavailability [12]. After ingestion, they are hydrolyzed in the upper part of the intestine and only a small portion is absorbed [13]. The rest is processed in the large intestine by the intestinal microbiota into low molecular weight fatty acids [14]. Hydroxytyrosol is formed by the hydrolysis of oleuropein during the maturation of olive oil [15]. Since it is a polar compound, it is absorbed by passive transport in the intestine [16], with an absorption rate above 40% [15]. Studies have shown that hydroxytyrosol is more effective when administered as EVOO than in aqueous solution since the antioxidants contained in EVOO protect hydroxytyrosol from degradation in the gastrointestinal tract [17,18]. In addition, it has recently been demonstrated that female animals can metabolize and utilize hydroxytyrosol more efficiently [19]. For oleuropein, the most abundant phenolic compound in olive leaves, and the seeds, pulp, and peel of unripe olives [2], the study showed greater stability during digestion and higher bioavailability compared to hydroxytyrosol. Oleuropein reached the colon unchanged and therefore produced more diverse microbial metabolites. Considering the bioavailability results, it appears that oleuropein may be the most suitable hydroxytyrosol precursor for inclusion in food or nutraceutical formulations [20]. However, when phenols are consumed as a combination in a food or a crude extract, they may have a greater impact on health than a purified single compound [2].

Among the many beneficial functions of olive polyphenols, the neuroprotective function is one of the most important. A particularly important effect is the reduction of neuroinflammation as this is a component of many neurodegenerative and demyelinating diseases such as multiple sclerosis (MS) and pro-stroke disorders [21]. MS is the most common of the demyelinating diseases and is very challenging to treat. It is a chronic, autoimmune, inflammatory, demyelinating, and neurodegenerative disease of the central nervous system (CNS) that leads to disability due to permanent damage or deterioration of the nerves [22]. There is no cure, but the usual treatment helps to accelerate recovery from relapses and alleviate symptoms. By counteracting inflammation and oxidative stress that lead to the destruction of myelin and neurons, olive polyphenols could be a very helpful adjuvant therapy to slow the progression of demyelination and neurodegeneration. In this review the potential mechanisms of their anti-demyelination action are discussed. Each of the aforementioned olive polyphenols is believed to have some degree of anti-inflammatory or antioxidant activity [6] (Figure 1).

Stroke is another neurological disease with challenging treatment options. It is the second leading cause of disability and death in adults aged 65 years and older worldwide [23]. A number of pro-stroke mechanisms or risk factors are involved in the pathogenesis of stroke, including hypertension, atherosclerotic processes, and pro-atherosclerotic conditions such as lipid profile disorders, hyperglycemia, endothelial dysfunction, oxidative stress and inflammation, insulin resistance, and platelet activation (Figure 2). Thus, to reduce the risk of stroke, action is needed on many fronts [24]. Data from a number of studies suggest that olive polyphenols reduce the risk of stroke due to their multiple anti-stroke effects, such as anti-atherosclerotic, antihypertensive, antioxidant, anti-inflammatory, hypocholesterolemic, hypoglycemic, and anti-thrombotic [25,26,27,28,29].

This review provides a unique approach to detailing the pathophysiological mechanisms underlying cerebrovascular and demyelinating diseases such as MS and cerebral insult. They are a major cause of morbidity, death, long-term disability, and social and economic burden, and a major challenge for medical treatment. Therefore, the review focuses on the beneficial anti-demyelinating and anti-stroke properties and mechanisms of action of olive polyphenols in the prevention/counteraction of these neurological diseases.

## 2. Methods

The Pubmed database was searched up to September 2022 using the keywords “hypertension”, “hyperlipidemia”, “hyperglycemia”, “insulin resistance”, “metabolic syndrome”, “platelet aggregation”, “stroke”, “coagulation”, “multiple sclerosis”, “oxidative stress”, “atherosclerosis”, “endothelial dysfunction”, “cerebrovascular disease”, “vascular dysfunction”, “inflammation”, “immune cells”, “microglia”, “cytokines”, “demyelination”, “iron”, “free radicals scavenging” in combination with “olive polyphenols”, “olive oil”, “olive leaf extract”, “oleocanthal”, “tyrosol”, “hydroxytyrosol”, “oleuropein”, “oleacein”. We also examined the reference lists of the retrieved articles. Criteria for inclusion in the review were as follows: (1) English language full-length publication in a peer-reviewed journal; (2) articles that directly addressed the topic of olive derivatives and their polyphenols.

## 3. Anti-Demyelination Action of Olive Polyphenols

Demyelinating diseases are characterized by the destruction of myelin leading to axon degeneration and neuronal loss, resulting in various neurological impairments [30]. The main representative of primary demyelination, caused by acute inflammatory damage to oligodendrocytes and myelin, is MS [31]. The disease usually affects younger people (between 20 and 40 years of age), and it is a major cause of disability in young adults and therefore a significant socioeconomic problem [32]. Neuroinflammation is an underlying mechanism of many neurodegenerative diseases, especially demyelinating diseases such as MS. It affects the CNS and indicates the presence of specific CNS autoantigens that trigger an immune response and cause tissue damage. Changes in the tissue promote the immune response and activation of microglia, which secrete cytokines and free radicals, leading to a local increase in oxidative stress and additional tissue damage [33]. MS is considered a T cell-mediated autoimmune disease. Myelin-reactive T cells migrate from the periphery across the blood–brain barrier (BBB) into the CNS, causing inflammation, demyelination, and neurodegeneration [34].

In addition to immunomodulation therapy used in patients with MS, natural compounds that can slow disease progression have been sought. Among others, EVOO and olive-derived polyphenols have been identified as compounds with anti-inflammatory activity. For a long time, it was assumed that the anti-inflammatory effects of EVOO were due to monounsaturated fatty acids, but recent studies have shown that polyphenols also play an important role [35]. What is known today is that polyphenols reduce morbidity and slow down the progression of neurodegenerative diseases [36]. They reduce oxidative stress and inflammation and modulate the immune system by affecting cytokine production and white blood cells’ activity [35].

Hydroxytyrosol and oleuropein seem to have the best neuroprotective properties [12]. To achieve their neuroprotective effect, polyphenolic compounds must pass through the BBB and reach an appropriate concentration in the brain [37]. The selective nature of the BBB determines the availability of polyphenols in the CNS. In general, it is known that the aglycones of polyphenols can be better absorbed because they can cross membranes by a passive diffusion [9].

There are several mechanisms that have been proposed for the anti-inflammatory and neuroprotective effects of polyphenols: the prevention of nuclear translocation and reduction of nuclear factor kappa B (NF-κB) transcription factor activity; the reduction of production of pro-inflammatory factors such as interleukin-1-β (IL-1β) and tumor necrosis factor-α (TNF-α); the downregulation of inducible nitric oxide synthase (iNOS) involved in the immune response that produces NO as an immune defense mechanism, i.e., a free radical with an unpaired electron; the downregulation of cyclooxygenase-2 (COX-2) and nicotinamide adenine dinucleotide phosphate (NADPH) oxidase expression; the induction of sirtulin-1/AMP-activated protein kinase (SIRT1/AMPK) activation that reduces microglial activation; the suppression of the suppressor of cytokine signaling proteins (SOCS) and the Janus kinase (JAK) signal transducer and activator of the transcription (STAT) signaling pathway of cytokine signaling; and the increase in the mRNA and protein levels of anti-inflammatory cytokine IL-10 [12,38].

### 3.1. Olive Polyphenols and Pro-Inflammatory Cytokine Release

NF-κB is a transcription factor that plays an important role in the regulation and production of pro-inflammatory cytokines [8]. NF-κB can be activated via the canonical/classical pathway, the non-canonical/alternative pathway, and the atypical pathway. Triggers for NF-κB stimulation are numerous, including microorganisms, pro-inflammatory cytokines, oxidative stress, modified proteins, and chemical agents [39]. Upon activation, NF-κB dimers translocate from the cytoplasm to the nucleus and regulate the expression of target genes such as IL1-β, TNF-α, and iNOS, leading to the release of pro-inflammatory cytokines [40]. It was found that oleuropein exerts its anti-inflammatory effect by inhibiting NF-κB activation and its translocation to the nucleus. It also inhibits COX-2, caspase-3, and iNOS [41], thereby reducing the release of key inflammatory cytokines. In MS NF-κB regulates the activity of various inflammatory cells (T cells, macrophages, and microglia) and the viability of resident cells in inflammatory lesions. NF-κB plays an important role in T cell development and differentiation, and NF-κB activation in T cells promotes the development of MS. Interestingly, in MS, according to the available evidence, NF-κB activation in inflammatory cells enhances inflammation, while in oligodendrocytes and neurons, it protects these cells from inflammation, showing the double-edged nature [42].

An in vitro study [43] on BV-2 microglial cells showed that oleuropein suppressed the lipopolysaccharide (LPS)-induced increase in pro-inflammatory mediators and pro-inflammatory cytokines by inhibiting the activation of extracellular signal-regulated kinase (ERK)/p38/NF-κB, supporting a potential role of oleuropein in microglial inflammation-mediated neurodegenerative diseases. Furthermore, using an in vivo rat model, Shang et al. [44] demonstrated a significant decrease in IL-1β and TNF-α levels via the PI3K/Akt/mTOR signaling pathway, which is involved in the regulation of pro-inflammatory NF-κB activity.

### 3.2. Olive Polyphenols and Immune Cells

It is believed that MS is caused by a T cell-mediated autoimmune response [45]. T lymphocytes, along with B lymphocytes, are major components of the adaptive immune system [46]. They are involved in cytokine production, cytotoxicity, and the activation of other immune cells [47]. Depending on the expression of CD4 or CD8 molecules, T cells can be divided into CD8+ cytotoxic T cells, CD4+ T helper (Th) cells, and CD4+ Foxp3+ regulatory T cells (Treg) [48]. CD4+ Th cells can differentiate into different subtypes, such as Th1, Th2, and Th17 [49]. Th1 and Th17 phenotypes promote autoimmune tissue damage, whereas the Th2 phenotype is associated with attenuation of the immune response [30]. 

Evidence highlights an important role of CNS antigen-specific CD4+ T cells in the initiation of MS, while CNS antigen-specific CD8+ T cells cause CNS damage during relapses and in the chronic phase of the disease [50]. Since axons and neurons express only MHC class I, they can be directly damaged by CD8+ cells, whereas CD4+ T cells can only cause damage through indirect mechanisms [51].

Miljkovic et al. [52] demonstrated the efficacy of oleuropein in experimental autoimmune encephalomyelitis, an animal model of MS. Its application resulted in the downregulation of interferon-γ (IFN-γ) and IL-17 production during disease progression, counteracting pathogenic autoimmune responses that depend on Th17-, IFN-γ-, and IL-17-producing cells. In another study, the immunomodulatory effect of oleuropein was demonstrated with the suppression of T lymphocyte proliferation and a decrease in the autoimmune inflammatory process [53].

Neuroinflammation is mainly characterized and caused by activated microglial cells. Microglia are resident cells in the brain and are the first line of defense [12]. They perform different functions in the CNS and exhibit different phenotypes accordingly. The classical classification of microglial cells is into the pro-inflammatory iNOS-positive M1 and anti-inflammatory arginase-1-positive M2 phenotypes, but when stimulated with different substances, they can express a whole spectrum of phenotypes [54]. M1 and M2, as well as microglia, can transit from one phenotype to another. M2 microglia, with the help of arginase-1, switch arginine metabolism to ornithine and increase the production of polyamines, which are important for the extracellular matrix and collagen synthesis and contribute to the trophic support of neurons [12,55]. In addition, M2-activated microglia can repair minor damage, degrade toxic aggregates, and produce anti-inflammatory interleukins [55]. Therefore, shifting microglial cells from the M1 to M2 phenotype promotes neuroregeneration, has anti-inflammatory and neuroprotective effects [56], and may be a promising therapeutic target for neurodegenerative diseases [57]. In vitro studies on microglial cells have shown that hydroxytyrosol can alter the activation status of microglia by decreasing M1 markers and pro-inflammatory cytokine production and increasing M2 markers [55].

Microglia are activated during MS by Th1 cytokines produced by myelin-reactive T cells. This leads to microglial production of various inflammatory mediators and the enhancement of inflammation and CNS neurodegeneration [58]. It was found that oleuropein significantly attenuates microglial activation by acting on NF-κB activation [12]. 

Taticchi et al. [59] investigated the anti-inflammatory effect of virgin olive oil phenolic extracts with LPS-stimulated BV2 microglia. The results showed that phenolic extracts attenuated toll-like receptor 4 (TLR4) and NLRP3 signaling activation in LPS-stimulated microglial cells.

### 3.3. Olive Polyphenols and Oxidative Stress

Oxidative stress represents an imbalance between the production of free radicals and their scavenging. The result is the formation of reactive oxygen species (ROS) and oxidative damage to tissue [60]. Brain tissue has several characteristics that make it vulnerable to oxidative stress. It has a high-energy demand and therefore a high-energy consumption and metabolic rate. In addition, brain tissue is rich in polyunsaturated fatty acids (PUFA) and has low antioxidant defenses [9]. The result is oxidative stress, which is a common feature of various neurodegenerative diseases [60].

Despite the still unresolved etiology, it is known that neuroinflammation is the basis of the demyelination process in MS [61]. Inflammation leads to the formation of oxygen radicals and oxidative stress. Oxidative stress leads to further demyelination and neurodegeneration in two ways. First, through direct oxidation of lipids leading to lipid peroxidation and demyelination, and second, indirectly through the induction of a pro-inflammatory immune response [62]. An important source of ROS could also be impaired mitochondrial function, as has been reported in several neurodegenerative disorders, including MS [63].

Many reports have shown that EVOO polyphenols have antioxidant properties and may act against oxidative stress in brain tissues through various mechanisms. For example, oleuropein and hydroxytyrosol act as free radical scavengers, while hydroxytyrosol and oleocanthal are potent COX inhibitors. In addition, oleuropein counteracts the oxidation of low-density lipoprotein (LDL) [64,65]. Oleuropein exhibits potent antioxidant activity mainly by scavenging free radicals, as oleuropein can donate the electron [66]. In fact, in the chemical structure of oleuropein there are hydroxyl groups that are hydrogen donors and thus protect against oxidation. Its effects include increasing the antioxidant capacity of the brain and reducing the release of pro-inflammatory cytokines and chemokines, thus preventing neuroinflammation [67]. Several in vitro studies [68,69,70] confirmed that oleuropein suppresses the production of reactive oxygen and nitrogen species, reduces the release of myeloperoxidase by human neutrophils, and thus reduces the formation of hypochlorous acid. In addition, an animal study showed that oleuropein inhibits superoxide and nitric oxide (NO) production, decreases lipid peroxidation, and increases superoxide dismutase (SOD) activity [8]. Tyrosol does not have antioxidant activity that is as potent as oleuropein, but it is still very effective because of its high intracellular accumulation [71].

The studies on LPS-induced BV-2 microglial cells have shown that oleuropein suppresses the formation of a ROS. Namely, by inhibiting LPS-induced dynamin-related protein-1 (Drp-1)-dependent mitochondrial fission, oleuropein suppresses the excessive mitochondrial fission responsible for mitochondrial ROS generation and consequently suppresses the pro-inflammatory response of microglia [41].

NADPH oxidase, together with iNOS, is considered the most important enzyme in the production of ROS during inflammation [71]. It is required for the production of ROS in activated microglia. During activation of NADPH, the cytosolic subunits translocate to the membrane-binding cytochrome to form the oxidase that catalyzes the reduction of oxygen to superoxide free radicals. Hydroxytyrosol and its derivatives inhibit the expression of NADPH oxidase, thus inhibiting the production of ROS and the progression of inflammation [72].

A complex pathomechanism leading to demyelination and neurodegeneration in MS also involves iron in the form of iron deposition in the brain. Through the Fe-dependent conversion of superoxide anions and hydrogen peroxide into the extremely reactive hydroxyl radicals [73,74], iron can trigger cell death [75] and, subsequently, the propagation of demyelination and neurodegeneration. Moreover, iron released from dying oligodendrocytes and, subsequently, accumulated in microglia, astrocytes, or axons, can cause an amplification of the inflammatory oxidative burst in the CNS [76]. A murine animal model showed that hydroxytyrosol counteracts Fe^2+^- and NO-induced loss of cellular ATP and the depolarization of mitochondrial membrane potential in brain cells [77]. Additionally, due to its metal-chelating activity, it is able to protect membranes from lipid peroxidation induced by metals, including Fe [8]. Thus, the lowering of endogenous iron stores or the regulation of pro and antioxidant enzymes by olive oil polyphenols could be involved in the important mechanisms against demyelination [12,78,79].

## 4. Anti-Stroke Action of Olive Polyphenols

Of all neurological disorders, cerebrovascular diseases are responsible for more than half of the burden and 85% of the deaths related to neurological diseases worldwide. Stroke was the second leading cause of death, for about 143 million people, worldwide in 2019, with numbers expected to increase by 2030 [23]. Here the anti-stroke actions of olive polyphenols in pro-stroke conditions such as hyperglycemia, hyperlipidemia, oxidative stress, atherosclerosis, cerebral ischemia, and diastolic and systolic hypertension are discussed.

### 4.1. Olive Polyphenols and Hypertension

Hypertension is one of the most common disorders and, more importantly, is the main risk factor for cerebral insult. Due to high intraluminal pressure, the endothelium and smooth muscle in intracerebral arteries are exposed to increased stress leading to structural changes within the vessel wall. This leads not only to local thrombus formation, stenosis, occlusion, and ischemic lesions but also to the arteriosclerotic process. In other words, it leads to degenerative changes in smooth muscle cells and the endothelium, resulting in increased arterial stiffness associated with an increased risk of stroke [80,81]. Furthermore, all these changes lead to endothelial dysfunction and inadequate compensatory mechanisms unable to protect microvessels from increased pressure [82]. This also accelerates ischemic attacks and cerebral infarcts or predisposes the subject to plasma extravasation and focal cerebral edema, lacunar infarcts, and intracerebral hemorrhage [83,84].

The vascular endothelium plays an important role in regulating vascular tone through the synthesis of NO, prostaglandins, and other relaxing factors and provides protection against oxidative, inflammatory, thrombotic, and atherosclerotic processes, thus serving as a controller of normal blood pressure [85,86]. The essential homeostatic processes regulated by the endothelium are modulated by crosstalk between endothelial cells and other vascular cell types, including smooth muscle cells, monocytes, and macrophages, which contribute to normal vascular function. However, impaired communication between endothelial cells and these vascular cell types has been associated with vascular dysfunction and pathological remodeling in hypertension, atherosclerosis, atherothrombosis, and other cardiovascular diseases [87].

Endothelial dysfunction may also precede the development of hypertension by contributing to increased peripheral resistance, i.e., through activation of the renin–angiotensin system, endothelin-1, catecholamines, and growth factors’ production [82,88]. In addition, the induction of inflammatory processes and the production of ROS in the vessel wall may be associated with endothelial dysfunction. ROS are key signaling molecules through which vasoactive agents such as angiotensin II, endothelin-1, and others modify cell function through highly-regulated, redox-sensitive signal transduction. ROS stimulate multiple signaling pathways involved in inflammation and vascular remodeling (NF-κB, MAPK, JAK-2, STAT, etc.) and thus are involved in the development of hypertension [85].

There are many studied antihypertensive mechanisms of olive polyphenols, and yet this is only the beginning of a larger picture. Sustained consumption of phenolic-enriched virgin olive oil with different polyphenols causes improvements in endothelial function in humans [89]. The antihypertensive effects of olive oil and olive leaves are mainly attributed to the main phenolic compounds that reduce blood pressure through a number of specific mechanisms. Oleuropein is the most commonly described polyphenolic compound. A large number of studies in humans and laboratory rats have demonstrated its antihypertensive activity [90,91]. Olive leaf polyphenols have been shown to lower diastolic and systolic blood pressure in prehypertensive and hypertensive groups of patients [92,93,94,95,96,97].

The beneficial effects that olive oil and leaf phenolic compounds may have on endothelial function include the inhibition of monocyte adhesion and platelet activation, improvement in vasodilation [25] through the modulation of potent vasodilator and vasoconstrictor agents such as NO and endothelin-1 (ET-1) [98], and, furthermore, through the expression of their antioxidant and anti-inflammatory activity [97,99]. As stated earlier, all of these mechanisms of action are involved in the development of hypertension if not prevented or treated beforehand. The best known anti-inflammatory effects of olive polyphenols that can prevent endothelial dysfunction and progressive vasoconstriction include the inhibition of transcription factors NF-kB and AP-1, and the reduction in vascular TLR4 expression by the inhibition of mitogen-activated protein kinase (MAPK) signaling with the subsequent reduction in pro-inflammatory cytokines (IL-6, IL-10, TNF-α, IL-1β, etc.) and inflammatory markers (COX-2, PGE2, etc.) [6,100,101,102]. The novel anti-inflammatory effects of olive polyphenols hydroxytyrosol and oleuropein are thought to result from the suppression of phosphorylation and release of platelet heat shock protein 27, an extracellular pro-inflammatory agent [103]. As mentioned earlier, the antioxidant effects of olive polyphenols include upregulation of the expression of antioxidant enzymes (glutathione peroxidase (GPx), glutathione disulfide reductase (GSR), SOD-1, etc.), the elimination of elevated superoxide levels, the reduction in increased NADPH oxidase activity, and the activation of antioxidant cell proteins such as SIRT1 and peroxisome proliferator-activated receptor-γ coactivator-1α (PGC-1α) [25,104,105,106]. Moreover, olive leaf polyphenolic compounds oleuropein and hydroxytyrosol are efficient cytoprotective agents against H_2_O_2_-induced oxidative stress and toxicity in human umbilical vein endothelial cells [107].

In a study by Choi et al. [108], treatment with oleuropein significantly increased an angiotensin II-induced decrease in the levels of peroxiredoxin (Prdx)-1 and -2 in vascular progenitor cells, which have an important antioxidant and protective function in cells. By regulating the expression of Prdx-1 and Prdx-2 and by activating the ERK1/2 phosphorylation cascade, oleuropein decreases cellular ROS levels and reduces oxidative stress. Hydroxytyrosol and tyrosol from olive mill wastewater, a by-product of olive oil processing, also showed potent antioxidant capabilities to counteract H_2_O_2_-induced oxidative stress and cell death in cell viability experiments that included endothelial cells and vascular smooth muscle cells [109]. 

In animal models of simultaneous renal hypertension and type 2 diabetes associated with the impaired release of NO, oleuropein showed sympathoplegic activities with subsequent lowering of systolic blood pressure. In addition to lowering systolic blood pressure, it also decreased the maximal response to phenylephrine and increased the maximal response to acetylcholine [110].

The antihypertensive effects of oleuropein can be seen by its negative chronotropic and inotropic effects on the heart [111]. This effect could be of great importance in isolated systolic hypertension. Moreover, virgin olive oil has been shown to have an inhibitory effect on angiotensin-converting enzyme (ACE) and thus to have a vasodilatory effect in spontaneously hypertensive rats [112,113]. Overall, olive polyphenols have a beneficial effect on cardiovascular parameters, including sequels of hypertensive disorder such as heart failure, myocardial infarction, and renal hypertension [25,26,27].

Many studies have confirmed the endothelium-dependent vasorelaxant effect of olive polyphenols. As previously mentioned, NO is an important protective molecule in the vascular system, and endothelial NO synthase (eNOS) is responsible for most of the vasodilatory NO production [114]. Olive phenolic compounds perform a particular task to increase NO bioavailability and the expression of eNOS [115]. an imbalance between NO and ET-1 leads to endothelial dysfunction, and it is possible that hyperglycemia and hyperlipidemia may decrease endothelial NO synthase phosphorylation and, consequently, intracellular NO levels and increase ET-1 synthesis. Hydroxytyrosol and extra virgin olive oil polyphenol extract partially reverse these abnormalities [98]. In addition, it was observed that hyperglycemia and free fatty acids decreased NO and increased acetylcholine-induced ET-1 levels by modulating intracellular calcium concentrations and endothelial NO synthase phosphorylation, events that were also reversed by hydroxytyrosol and the polyphenol extract [98]. Moreover, simple phenols from extra virgin olive oil, tyrosol, and hydroxytyrosol can modulate the NO balance by decreasing its degradation (through decreased superoxide formation) and increasing its production through the Akt1/eNOS pathway [116].

Another possible mechanism by which oleuropein exerts the antihypertensive effect is by affecting renal water reabsorption in renal cells, i.e., by preventing vasopressin-induced aquaporin-2 translocation to the plasma membrane of renal cells [117]. In addition to diuresis, it also has a stimulating effect on natriuresis [118].

### 4.2. Olive Polyphenols and Vascular Dysfunction

Vascular dysfunction includes large artery dysfunction due to arterial stiffness, microvascular dysfunction (dysfunction of the microcirculation), and endothelial dysfunction (dysfunction of the endothelium) [119]. Arterial wall stiffening is determined by the reduced elastin–collagen ratio caused by the production of ROS and by inflammation, aging, hypertension, hyperglycemia, and dyslipidemia; all mechanisms that olive polyphenols reduce and counteract. In addition, arterial stiffness due to decreased compliance of the large peripheral arteries leads to increased pulsatile pressure and blood flow stress, which damage the cerebral microcirculation. Cerebral capillaries are particularly vulnerable to this damage due to low impedance, which can lead to a reactive increase in vascular resistance and subsequent impaired vasoreactivity and microvascular ischemia [120]. Olive polyphenol ingestion may help prevent these cerebral disorders, and thus cerebral stroke, by reducing blood pressure and associated endothelial dysfunction, and the occurrence of arterial stiffness. Arterial stiffness and endothelial dysfunction are distinct aspects of vascular disease, but there is certainly a crosstalk between these two pathophysiological processes [121]. As mentioned earlier, endothelial dysfunction is characterized by an imbalance between vasoconstrictor and vasodilator factors, by increased levels of ROS and pro-inflammatory factors, and by decreased NO secretion. All these lead to increased vasoconstriction, leukocyte adhesion, smooth muscle cell proliferation, extracellular matrix deposition, cell adhesion, platelet activation, prooxidation, thrombosis, impaired coagulation, vascular inflammation, and atherosclerosis [86,122]. In the previous section, the mechanisms of the antihypertensive action of olive polyphenols and their beneficial role in combating endothelial dysfunctionality were presented. Here the anti-atherosclerotic properties and mechanism of action of olive polyphenols, i.e., against increased lipoprotein oxidation, smooth muscle cell proliferation, extracellular matrix deposition, hyperglycemia, hyperlipidemia or hypercholesterolemia, and thrombus formation (platelet activation), all of which are closely associated with the atherosclerosis process, are discussed.

#### 4.2.1. Olive Polyphenols and Atherosclerosis

Atherosclerosis is mainly caused by the accumulation of cholesterol, fat, and other compounds in or on the arterial walls, leading to the restriction of blood flow to the organs [123]. Elevated LDL cholesterol levels are directly related to the risk of atherosclerotic cardiovascular events. The infiltration and retention of LDL cholesterol in the arterial wall triggers an inflammatory response and promotes the development of atherosclerosis. Arterial injury leads to dysfunction of the endothelium, which promotes the modification of lipoproteins and infiltration of monocytes into the subendothelial space. Macrophage inflammation results in enhanced oxidative stress and cytokine/chemokine release, which in turn causes increased LDL oxidation, endothelial cell activation, monocyte recruitment, and foam cell formation. High-density lipoprotein (HDL) prevents inflammation and oxidative stress and promotes cholesterol efflux to reduce lesion formation [124]. Oxidative stress and ROS have been suggested to play a significant role in endothelial dysfunction and subsequent atherosclerosis pathogenesis [125]. Overproduction of ROS facilitates the oxidative modification of LDL and phospholipids, reduction in NOS-derived nitric oxide, and functional disruption of HDL, which are profoundly involved in atherogenesis. It has been widely studied as to whether polyphenolic compounds from olive oil and leaves can reduce or reverse endothelial damage induced by oxidation and inflammation [101,109,125].

In addition, many studies have reported that polyphenol-rich olive oil or leaves have anti-atherosclerotic properties by preventing cholesterol accumulation in macrophages through the suppression of lipoprotein oxidation and the regulation of cholesterol uptake and efflux [126]. Consumption of polyphenol-rich olive oil reduces systemic LDL oxidation and pro-atherogenic genes in peripheral blood mononuclear cells, i.e., CD40-ligand, IL-23α, adrenergic β-2 receptor (ADRB2), oxidized LDL (lectin-like) receptor 1 (OLR1), and IL-8 receptor-α (IL8RA) in humans [127]. Moreover, hydroxytyrosol has an anti-atherogenic effect by preventing the lipid peroxidation process, i.e., by upregulating the antioxidant enzymes SOD and catalase in the liver [128]. In addition, there is convincing evidence that olive polyphenols reduce LDL oxidation and improve blood glucose and blood pressure control [129]. The atheroprotective effects of extra virgin olive oil polyphenols on HDL-mediated cholesterol efflux and oxidative stress damage, including tyrosol and hydroxytyrosol, appear to be mediated by a signaling mechanism that acts either independently or synergistically. They stimulate the expression of ATP-binding cassette transporter A1 (ABCA1) protein in macrophages, one of the major cholesterol efflux pathways that plays an important role in maintaining cellular cholesterol homeostasis by participating in the reverse cholesterol transport pathway. Thus, they enhance the anti-atherogenic properties of HDL by reducing the oxidative modifications to HDL and maintaining the physicochemical properties of HDL, which in turn improves the functionality of HDL [130]. It was found that oleacein inhibits the formation of foam cells in human monocyte-derived macrophages, making it a valuable agent for the prevention of early and advanced atherosclerosis [131]. Oleacein significantly reduces lipid deposition in macrophages as well as their expression of selected scavenger receptors in a dose-dependent manner. The possible mechanism of active cholesterol efflux from macrophage foam cells induced by oleacein involves the upregulation of key cholesterol efflux pathways, i.e., scavenger receptor class B member 1 (SRB1), ABCA1, and ATP-binding cassette transporter G1 (ABCG1) [131]. Furthermore, oleacein significantly reduces early apoptosis of macrophages stimulated by oxidized low-density lipoprotein (oxLDL) and increases protein expression of the JAK/STAT3 pathway involved in inflammation and vascular remodeling and thus atherosclerosis [132]. Moreover, oleacein increases intracellular secretion of heme oxygenase-1 with important anti-inflammatory and antioxidant activities and almost completely inhibits the transcription factor NF-κB [131] leading to the regression of atherosclerotic plaques [133]. Oleacein has been shown to contribute to the inhibition of destabilization and rupture of carotid atherosclerotic plaques that could trigger embolism into the cerebral vasculature and cause ischemic cerebral insult [134]. In addition to the mentioned anti-atherogenic mechanisms, olive polyphenols may have anti-proliferative effects on smooth muscle cells. It has been shown that oleuropein, hydroxytyrosol, and tyrosol inhibit smooth muscle cell proliferation through cell cycle blockade, possibly regulated by ERK1/2 [102].

An in vitro study by Scoditti et al. [135] showed an inhibitory effect of oleuropein and hydroxytyrosol on COX-2 and the matrix-degrading enzyme matrix metalloproteinase-9 (MMP-9), which are known to contribute to inflammation, atherosclerotic lesion formation, and vulnerability.

Alterations in mitochondrial membrane potential induce superoxide production and lead to oxidative stress, which in turn leads to endothelial/vascular dysfunction. The antioxidant effect of hydroxytyrosol is associated with decreased mitochondrial superoxide production and lipid peroxidation and increased SOD activity [136]. In an in vitro model of endothelial dysfunction represented by cultured endothelial cells treated with phorbol myristate acetate (PMA = an inflammatory, prooxidant, and proangiogenic agent), hydroxytyrosol improves endothelial mitochondrial function by reversing the PMA-induced decrease in mitochondrial membrane potential, ATP synthesis, and ATP5β expression. Hydroxytyrosol also promotes mitochondrial biogenesis by increasing mitochondrial DNA content and mitochondrial biogenesis factor expression [136].

#### 4.2.2. Olive Polyphenols and Hyperlipidemia

As described in the previous section, hyperlipidemia is the most important risk factor for atherosclerosis, which is the major cause of cardiovascular disease. The accumulation of cholesterol in macrophages in the arterial intima is due to increased oxidation of LDL and their uptake by macrophages. Cholesterol efflux from cholesterol-laden macrophages into HDL is another important process to prevent cholesterol accumulation.

Many studies have reported that polyphenol-rich olive oil or leaves have anti-atherosclerotic properties by lowering LDL levels and increasing HDL levels [137]. Olive oil polyphenol consumption directly lowers LDL levels (i.e., decreases apo B-100 levels and total and small LDL particle numbers) and LDL atherogenicity (i.e., increases LDL resistance to oxidation) in healthy young people [138]. Moreover, olive leaf extract polyphenols significantly lower total cholesterol, total LDL cholesterol, total cholesterol/HDL cholesterol ratio, and oxidized LDL in hypercholesterolemic subjects [129,139]. Consumption of polyphenol-rich olive oil promotes the main anti-atherogenic function of HDL, its cholesterol efflux capacity. These polyphenols increase HDL size, promote greater HDL stability reflected in a low-triglyceride core, and improve HDL oxidative status through an increase in olive oil polyphenol metabolite lipoprotein content [140]. In the study by Knaub et al. [141], a significant change in LDL cholesterol levels was observed in the hydroxytyrosol group after a 12-week intervention, but there was no change in total cholesterol, HDL cholesterol, LDL/HDL ratio, and triglycerides. However, these results also suggest an anti-atherogenic effect [142]. On the other hand, when the same lipid parameters were measured in hyperlipemic rabbits, hydroxytyrosol decreased total cholesterol and triacylglycerols, increased HDL cholesterol, improved antioxidant status, and decreased atherosclerotic lesion size measured as the area of the intimal layer of the aortic arch [143]. This is a very important finding because olive polyphenols could be used not only to prevent atherosclerotic lesions but also to regress them.

A very important effect of olive oil or olive leaf polyphenols is to reduce the risk of metabolic syndrome by reducing visceral fat, which may improve blood lipid profiles and reduce the atherosclerotic process. In addition, consumption of extra virgin olive oil with a high concentration of the phenolic compound oleocanthal has beneficial effects on metabolic syndrome parameters, i.e., a reduction in abdominal fat, body weight, waist circumference, body mass index (BMI), pro-inflammatory cytokines, and fatty liver index in patients at high cerebrovascular risk [28].

It is well known that the inflammation and dysfunction of adipose tissue in obesity, leading to the excessive production of inflammatory mediators, are the main processes linking obesity to insulin resistance, type 2 diabetes, and cardiovascular disease [144]. EVOO polyphenols, oleocanthal and oleacein, significantly reduce the expression of genes involved in adipocyte inflammation (IL-1β, COX-2), angiogenesis (VEGF/KDR, MMP-2), oxidative stress (NADPH oxidase), antioxidant enzymes (SOD and GPX), chemotaxis, and the infiltration of leukocytes (MCP-1, CXCL-10, MCS-F), and enhance the expression of the anti-inflammatory/metabolic effector PPARγ. Accordingly, miR-155-5p, miR-34a-5p, and let-7c-5p, which are closely related to the NF-κB signaling pathway, are deregulated by TNF-α in both cells and exosomes [145]. Moreover, the virgin olive oil component hydroxytyrosol at nutritionally relevant concentrations has a beneficial effect in preventing adiponectin downregulation in inflamed adipocytes by attenuating JNK-mediated PPARγ suppression. Namely, adiponectin is an adipocyte-derived, insulin-sensitizing, and anti-inflammatory hormone that is suppressed in obesity by mechanisms related to chronic inflammation and oxidative stress [146]. Therefore, these compounds could be novel dietary agents for the prevention of obesity-related inflammatory diseases.

#### 4.2.3. Olive Polyphenols and Hyperglycemia

Hyperglycemia-induced endothelial dysfunction is the major factor in the development of vascular disease in diabetes mellitus [147,148]. Hyperglycemia and early disglycemia caused by obesity-related insulin resistance or impaired insulin secretion, cause several alterations at the cellular level of vascular tissue that may accelerate the atherosclerotic process: (1) glycosylation of proteins and lipids, which may affect their normal function by disrupting molecular conformation and enzymatic activity, reducing degradation capacity, and impairing receptor recognition. In addition, glycosylated proteins interact with monocyte-derived macrophages, endothelial cells, and smooth muscle cells in the atherosclerotic process. Interaction of glycosylated proteins with their receptor leads to the induction of oxidative stress and pro-inflammatory responses; (2) oxidative stress; (3) the activation of protein kinase C with the subsequent alteration of growth factor expression [149]. In addition, high glucose concentration over a prolonged period can lead to mitochondrial hyperpolarization [150].

Polyphenols from olive oil and leaves are thought to inhibit mitochondrion ROS and are therefore valuable agents in diabetes therapy [151]. Endothelial dysfunction is primarily responsible for impaired vasorelaxation and vasodilator factor production in diabetes, but it is closely followed by the development of vascular smooth muscle cell dysfunction [152,153]. The improvement in hyperglycemia with concomitant consumption of olive polyphenols can restore endothelium-dependent vasodilation in experimental models of diabetes [152]. Moreover, the improvement in insulin sensitivity by olive polyphenols is associated with the restoration of flow-mediated vasodilation [2,154].

In obese individuals, adipocyte-induced inflammation and macrophage activation are major causes of insulin resistance. The endothelium plays a central role in this process as the blockade of vascular inflammation and oxidative stress prevent insulin resistance and increase peripheral insulin sensitivity, respectively. These findings may be a promising approach to prevent metabolic disorders using polyphenol-rich olive products, including olive leaves and their crude extract, because of their anti-inflammatory and antioxidant effects [147,155,156]. In addition, olive polyphenols significantly lower blood glucose and triglycerides by upregulating GLUT4 expression, i.e., promoting translocation of GLUT4 to skeletal muscle. Rab8a, Rab13, and Rab14 partially contribute to GLUT4 translocation regulated by olive leaf polyphenols [79].

Human intervention involving the ingestion of a single dose of extra virgin olive oil enriched with phenolic compounds from the olive fruit improves postprandial insulin release and peripheral tissue sensitivity in healthy adults [29]. Furthermore, oleacein, one of the most abundant secoiridoids in EVOO, showed protective effects by improving body weight, insulin sensitivity, serum lipids, and liver tissue histology in mice [157]. Consumption of olive polyphenols, especially oleuropein, can significantly improve insulinemia and insulin degradation and also reduce β-cell apoptosis, increase β-cell number, and normalize islet glucose metabolism and glucose-induced insulin secretion [158,159,160,161].

Recent findings revealed another mechanism by which oleuropein may ameliorate type 2 diabetes. It modulates the composition and function of the gut microbiota, which is an important finding because modulation of the microbiota is important not only for improving diabetes but also possibly for other metabolic diseases [162]. A similar beneficial effect on the microbiota was found with hydroxytyrosol, another common compound of olive leaves [163,164].

#### 4.2.4. Olive Polyphenols and Thrombosis (Platelet Aggregation)

Endothelial cell and smooth muscle cell dysfunction are the main features of vasculopathy, favouring a pro-inflammatory/-thrombotic state that eventually leads to atherothrombosis. Platelets play a key role in the formation of vascular plugs and atherosclerotic plaques by acting as mediators of tissue homeostasis and may also modulate the microenvironment of the atherosclerotic plaque [82]. Normally, insulin inhibits platelet aggregation and thrombosis by inhibiting tissue factor [165] and increases fibrinolytic activity by modulating plasminogen activator inhibitor levels [166]. Hyperglycemia stimulates coagulation, and hyperinsulinemia impairs fibrinolysis [167]. In addition, insulin resistance promotes atherothrombosis through impaired fibrinolysis, possibly due to the increased thrombin-activatable fibrinolysis inhibitor [168,169].

Thus, regulation of hyperglycemia and insulin levels in the prediabetic and diabetic states by olive polyphenols might also reduce the pro-thrombotic risk for cerebral insult. The cellular and animal studies clearly show that the polyphenols of extra virgin olive oil are able to inhibit platelet activation [170,171]. Phenolic compounds extracted from olive oil and fractions derived from it inhibit platelet function and eicosanoid formation in vitro. They inhibit collagen-induced platelet aggregation and thromboxane B2 production [172].

Weekly intake of EVOO rich in oleocanthal also prevents platelet aggregation [173]. Olive oil polyphenols inhibit cAMP phosphodiesterase, an important enzyme for maintaining adequate platelet function [174]. A by-product of olive production, an alperujo extract containing the most polyphenols found in olive fruit, was found to have antiplatelet activity. It regulates platelet structure and aggregation by upregulating Rho GDP-dissociation inhibitor 1, a negative regulator of Rho family GTPases, which upregulates an anticoagulant protein annexin A5. Annexin A5 also regulates coagulation and plays an important role in the prevention of atherothrombosis [175]. Furthermore, alperujo extract decreases levels of integrin αIIb, which probably impairs the formation of the full integrin complex integrin αIIb/β3, also known as GPIIb/IIIa, a receptor found on platelets. It is a receptor for fibrinogen and von Willebrand factor and initiates platelet activation and platelet–platelet interaction [176].

Olive polyphenolic compounds are being investigated not only in their natural form as anti-aggregating agents, but also in sulfated form to serve as alternative anticoagulant therapeutics due to their higher bioavailability and degradation resistance compared to heparins. Tyrosol is an example that was recently studied using different sulfation approaches. Mild sulfation of a mixture of tyrosol oligomers was found to have the best anticoagulant properties, especially in terms of prolonging the activated partial thromboplastin time, with a good ability to further reduce thrombus [177].

As previously reported, moderate iron excess markedly accelerates thrombus formation, impairs vasoreactivity, and increases the production of ROS [73,74]. Moreover, the administration of ROS scavenger completely abolishes iron-induced thrombus formation, confirming that iron accelerates thrombosis through a prooxidant mechanism [178]. A reduction in endogenous iron stores induced by olive oil polyphenols may be involved in the beneficial effects of stroke prevention. In addition, Samieri et al. [78] reported that a higher consumption of olive oil with highly bioactive constituents such as polyphenols was associated with a lower incidence of stroke. This is likely due to the fact that prolonged consumption of olive polyphenols gradually decreases body iron stores and ferritin levels [179,180]. Moreover, a strong association between body iron stores and asymptomatic atherosclerosis has recently been demonstrated [181,182].

## 5. Conclusions

The beneficial effects of polyphenols from olive derivatives on neurological demyelinating and cerebrovascular disorders have been extensively studied and associated with the modulation of many different cellular pathways. They are studied as a combination in olive oil and leaves, leaf extract or other olive by-products, or as individual compounds such as oleocanthal, oleuropein, hydroxytyrosol, and tyrosol. 

In the context of reducing the risk of multiple sclerosis, olive polyphenols reduce neuroinflammation by decreasing oxidative stress through the reduction of ROS and upregulation of antioxidant enzymes, by the downregulation of pro-inflammatory transcription factors such as NF-kB and AP-1, by regulation of the secretion of pro- and anti-inflammatory cytokines, and by the production of pro-aggressive and regulatory T lymphocytes, and pro-inflammatory and anti-inflammatory microglia.

In the context of reducing the risk of cerebral insult, olive polyphenols improve endothelial function by acting directly on the vascular wall or indirectly through the entire system. They decrease oxidative stress and reduce inflammation, reduce and prevent atherosclerotic, atherothrombotic, and arteriosclerotic processes, and improve blood pressure, lipid profile, hyperglycemia, and peripheral insulin sensitivity.

Thanks to their multiple modes of action, olive polyphenols have a great potential for therapeutic success in combating multifactorial pathologies such as neurological disorders including multiple sclerosis and stroke. Therefore, further clinical trials are needed to gain more precise knowledge about preventive/therapeutic relevant concentrations and dosages.

## Figures and Tables

**Figure 1 nutrients-14-04533-f001:**
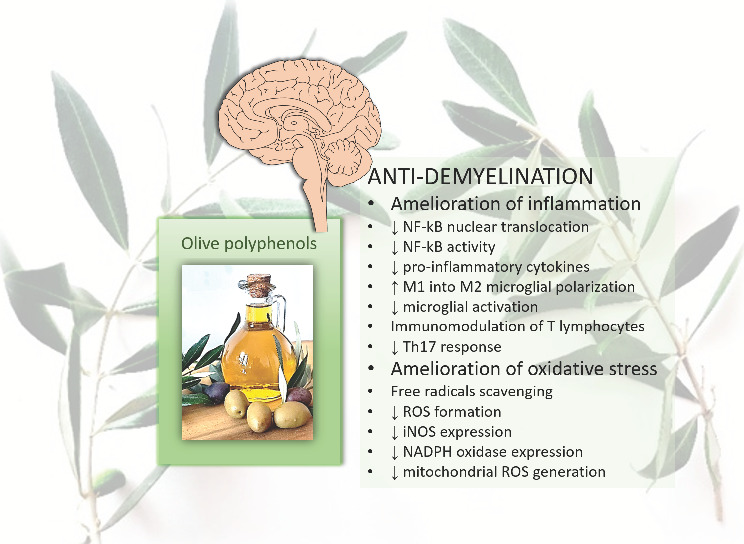
Main anti-demyelination effects of olive polyphenols. ↑, increases; ↓, decreases; NF-kB, nuclear factor kappa B; M1, microglia type 1; M2, microglia type 2; Th17, T helper 17 cells; ROS, reactive oxygen species; iNOS, inducible nitric oxide synthase; NADPH, nicotinamide adenine dinucleotide phosphate.

**Figure 2 nutrients-14-04533-f002:**
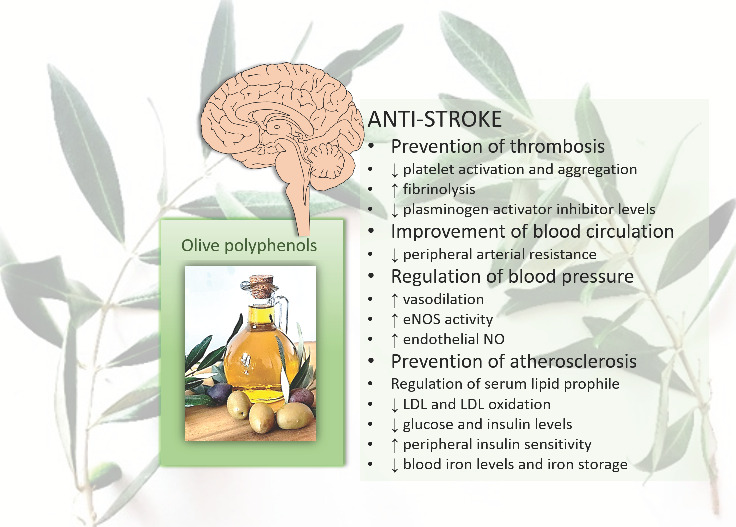
Main anti-stroke effects of olive polyphenols. ↑, increases; ↓, decreases; eNOS, endothelial nitric oxide synthase; NO, nitric oxide; LDL, low-density lipoprotein.

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
