# Peer review of "Neuroprotective Panel of Olive Polyphenols: Mechanisms of Action, Anti-Demyelination, and Anti-Stroke Properties"

_nutrients, 2022, doi:10.3390/nu14214533_

Round 1

Reviewer 1 Report

Kezelle & Ćurko-Cofek present a review manuscript on the neuroprotective mechanisms of polyphenols. Relative to a similar manuscript such as PMID 29068387 there are some recent studies cited and discussed among 2018 to 2022 with some additional perspective for vascular dysfunction as an example. The primary emphasis for disease models entails conditions of stroke and de-myelination with summary illustrations as Figures 1 & 2 respectively. See comments below.

(1) Major: Overall, it is difficult to see how the current review is substantially standing out as an original contribution to the literature relative to other numerous reviews on the same/similar topic. With just a brief peak at the literature, a reference such as PMID 29068387 is more comprehensive and organized for chemical structures and mechanistic & therapeutic value across disease states, albeit from 2017. The authors state the purpose of the review as an “updated” version but they should specify precisely how this manuscript builds from prior and ongoing work, ideally with unique perspective based on their own laboratory work.

(2) The manuscript needs another thorough editing. As one key example, there is a contradiction in Figure 2 for increased eNOS activity but decreased NO (?). In this context, the authors should specify the differences among NOS isoforms as eNOS, iNOS, and nNOS.   

Author Response

We thank the reviewer for the useful suggestions and effort he/she made to improve our manuscript, "Neuroprotective panel of olive polyphenols: anti-demyelination and anti-stroke properties and mechanisms of action". The responses and explanations to the comments are listed below and highlighted in the revised manuscript.

Reviewer 1   -   Comments to Authors:

Kezelle & Ćurko-Cofek present a review manuscript on the neuroprotective mechanisms of polyphenols. Relative to a similar manuscript such as PMID 29068387 there are some recent studies cited and discussed among 2018 to 2022 with some additional perspective for vascular dysfunction as an example. The primary emphasis for disease models entails conditions of stroke and de-myelination with summary illustrations as Figures 1 & 2 respectively. See comments below.

(1) Major: Overall, it is difficult to see how the current review is substantially standing out as an original contribution to the literature relative to other numerous reviews on the same/similar topic. With just a brief peak at the literature, a reference such as PMID 29068387 is more comprehensive and organized for chemical structures and mechanistic & therapeutic value across disease states, albeit from 2017. The authors state the purpose of the review as an “updated” version but they should specify precisely how this manuscript builds from prior and ongoing work, ideally with unique perspective based on their own laboratory work.

Response to the Reviewer:

We thank the reviewer for his/her comments.

Yes, we agree that the paper PMID 29068387 that we cited shows perhaps more detailed biochemical characteristics of polyphenols, but it speaks more generally about the beneficial effects of polyphenols on ischemic brain injury. It was not our intention to review the biochemical properties, pharmacokinetics, and pharmacodynamics of polyphenols precisely because so much has already been explained. In contrast, here in our paper, we describe in detail for the first time the key pathophysiological mechanisms of vascular dysfunction related to the specific problem of the major risk factors for stroke, hypertension, and vascular dysfunction associated with hyperlipidemia, hyperglycemia, endothelial dysfunction, arteriosclerosis, atherosclerosis, etc.). We explain in detail the pathophysiological effects of hypertension and link each part to the effects of polyphenols. We also do this with the pathophysiological mechanisms that lie in the background of the demyelination process, whereas the paper PMID 29068387 relates more to neurodegenerative diseases such as PD, AD, ALS and MS in general.

Here we present new findings from our personal scientific contribution:
https://doi.org/10.1155/2020/6125638
https://doi.org/10.3390/ijms21238981
https://doi.org/10.1016/j.neuro.2016.08.014

https://doi.org/10.1016/j.mehy.2017.07.022

Therefore, our work has a unique approach to the detailed pathophysiological mechanisms related to the problem of cerebrovascular and demyelinating diseases with an emphasis on the connection with the beneficial effect of polyphenols on the same pathophysiological mechanisms. Moreover, we consider such an approach extremely useful and one of a kind because the reader can find in one place detailed pathophysiological explanations of the main risk factors for stroke and the beneficial effects of polyphenols from olive tree derivatives and become more aware of his diet. The reader can also see which main pathophysiological effects underlying demyelination can be influenced by polyphenols from olive tree derivatives, supporting the slowing of the progression of MS. In this context we changed a bit the purpose of this review: Line 90-95.

(2) The manuscript needs another thorough editing. As one key example, there is a contradiction in Figure 2 for increased eNOS activity but decreased NO (?). In this context, the authors should specify the differences among NOS isoforms as eNOS, iNOS, and nNOS.

Response to the Reviewer:

We thank the reviewer very much for pointing out this important mistake that we have overlooked. In Figure 2, we have changed decreased NO to increased NO. We thank the reviewer for the suggestion to describe more in detail all isoformes of NOS enzyme. iNOS is mentioned in lines 268-269, and we added a short explanation of iNOS enzyme: Line 148-150. The role of eNOS was explained in lines 386-399.

We thank the editors and reviewers for their observations, suggestions, and valuable comments. We greatly appreciate it and sincerely hope that the editors and reviewers will be pleased with the revision of our manuscript.

Thank you for your consideration, we are looking forward to your answer.

Sincerely,

the authors.

Reviewer 2 Report

I suggest the following title: Neuroprotective panel of olive polyphenols: Mechanisms of action, anti-demyelination, and anti-stroke properties.

36-37. Choose to use or not abbreviations for all the polyphenol groups. It is a little bit confusing to read them mixed. 

87-91. You must show in the purpose that you search for the actions of polyphenols too. It is a part of your title. 

135. Reading the paragraph I clearly understand the beneficial role of polyphenols. However, it would be helpful to give some information about their intake quantity. This is advice for the whole manuscript. 

It would be helpful to add more tables or figures to depict your results. This will give more interest to the readers. 

394. Avoid 1st plural.

Author Response

We thank the reviewer for the useful suggestions and effort he/she made to improve our manuscript, "Neuroprotective panel of olive polyphenols: anti-demyelination and anti-stroke properties and mechanisms of action". The responses and explanations to the comments are listed below and highlighted in the revised manuscript.

Reviewer 2   -   Comments to Authors

I suggest the following title: Neuroprotective panel of olive polyphenols: Mechanisms of action, anti-demyelination, and anti-stroke properties.

Response to the Reviewer:

We thank the reviewer for the useful comment. We totally agree with the suggestion. We agree that this title is more appropriate.

36-37. Choose to use or not abbreviations for all the polyphenol groups. It is a little bit confusing to read them mixed.

Response to the Reviewer:

We thank the reviewer for the useful comment. We decided to use the whole word for all polyphenols so we deleted the abbs.

87-91. You must show in the purpose that you search for the actions of polyphenols too. It is a part of your title. 

Response to the Reviewer:

We thank the reviewer for the useful comment. We changed a bit the purpose as suggested: line 90-95 and added “mechanisms of action” in the abstract as well.

  1. Reading the paragraph I clearly understand the beneficial role of polyphenols. However, it would be helpful to give some information about their intake quantity. This is advice for the whole manuscript.

 Response to the Reviewer:

We thank the reviewer for the useful comment. However, we initially had the idea to provide information on the intake amount along with every explanation of the mechanism of action, but in the end, we decided to do it in a different type of review that focuses more on the exact intake amount, bioviability, and biodynamics. In this particular review, we wanted to focus more on the effects of olive polyphenols on the underlying pathophysiological mechanisms of demyelination and stroke.

It would be helpful to add more tables or figures to depict your results. This will give more interest to the readers. 

Response to the Reviewer:

We thank the reviewer for the useful comment. We provided a graphical representation of the paper as a self-explanatory image to appear alongside with the abstract appearing on the Table of Contents. We hope this will help attract more readers and increase the visibility of our
paper once it is published online.

  1. Avoid 1st plural.

Response to the Reviewer:

We thank the reviewer for the useful comment. We changed the plural: Line 77, 296, 425, 427

We thank the editors and reviewers for their observations, suggestions, and valuable comments. We greatly appreciate it and sincerely hope that the editors and reviewers will be pleased with the revision of our manuscript.

Thank you for your consideration, we are looking forward to your answer.

Sincerely,

the authors.

Reviewer 3 Report

In the presented manuscript, the Authors describe the beneficial anti-demyelination and anti-stroke properties of olive polyphenols in the prevention / counteraction of neurological diseases. The study is interesting. However, the work lacks details of the methodology for developing this literature review - what databases did the authors use? what years? It is very important.

It should be mentioned that the olive leaf is a traditional raw material used in phytotherapy (European Union herbal monograph on Olea europaea L., folium, https://www.ema.europa.eu/en/documents/herbal-monograph/final-european) -union-herbal-monograph-olea-europaea-l-folium-first-version_en.pdf). In addition, unfortunately, the authors limit themselves only to polyphenolic compounds. Olives also contain oleanolic acid from the triterpenes group. Oleanolic acid has a neuroprotective effect (Castellano JM, Garcia-Rodriguez S, Espinosa JM, Millan-Linares MC, Rada M, Perona JS. Oleanolic Acid Exerts a Neuroprotective Effect Against Microglial Cell Activation by Modulating Cytokine Release and Antioxidant Defense Systems. Biomolecules. 2019 Nov 1; 9 ( 11): 683.doi: 10.3390 / biom9110683). It is worth considering the synergy of polyphenols and triterpenes activity.

Minor editorial notes: References should be cited in accordance with the guidelines for the journal - lines: 157, 175, 202, 333, 473, 578.

Author Response

We thank the reviewer for the useful suggestions and effort he/she made to improve our manuscript, "Neuroprotective panel of olive polyphenols: anti-demyelination and anti-stroke properties and mechanisms of action". The responses and explanations to the comments are listed below and highlighted in the revised manuscript.

Reviewer 3   -   Comments to Authors

In the presented manuscript, the Authors describe the beneficial anti-demyelination and anti-stroke properties of olive polyphenols in the prevention / counteraction of neurological diseases. The study is interesting. However, the work lacks details of the methodology for developing this literature review - what databases did the authors use? what years? It is very important.

Response to the Reviewer:

We thank the reviewer for the useful comment. We added the methodology: Line 97-108.

It should be mentioned that the olive leaf is a traditional raw material used in phytotherapy (European Union herbal monograph on Olea europaea L., folium, https://www.ema.europa.eu/en/documents/herbal-monograph/final-european) -union-herbal-monograph-olea-europaea-l-folium-first-version_en.pdf).

Response to the Reviewer:

We thank the reviewer for the useful comment. We added a sentence and a new reference in the reference list (5): Line 32-33.

In addition, unfortunately, the authors limit themselves only to polyphenolic compounds. Olives also contain oleanolic acid from the triterpenes group. Oleanolic acid has a neuroprotective effect (Castellano JM, Garcia-Rodriguez S, Espinosa JM, Millan-Linares MC, Rada M, Perona JS. Oleanolic Acid Exerts a Neuroprotective Effect Against Microglial Cell Activation by Modulating Cytokine Release and Antioxidant Defense Systems. Biomolecules. 2019 Nov 1; 9 (11): 683.doi: 10.3390 / biom9110683). It is worth considering the synergy of polyphenols and triterpenes activity.

Response to the Reviewer:

We thank the reviewer for the useful comment. We know that oleanolic acid has neuroprotective effects, and many other olive compounds as well. Unfortunately, we have to stop somewhere, so we decided for now to describe these particular compounds for this review. However, we are writing a chapter that will include oleanic acid and other compounds, and will describe in more detail their neuroprotective role.

Minor editorial notes: References should be cited in accordance with the guidelines for the journal - lines: 157, 175, 202, 333, 473, 578.

Response to the Reviewer:

We thank the reviewer for pointing these mistakes. We made the corrections: Lines 179,198,226,363,508,615.

We thank the editors and reviewers for their observations, suggestions, and valuable comments. We greatly appreciate it and sincerely hope that the editors and reviewers will be pleased with the revision of our manuscript.

Thank you for your consideration, we are looking forward to your answer.

Sincerely,

the authors.

Reviewer 4 Report

Grubić Kezele et.al did a great job introducing different types of Polyphenols and their effects on anti-demyelination and anti-stroke properties. The review paper is well structured and well written and ready for publish.

Author Response

We thank the reviewer for the very nice comments to our manuscript titled "Neuroprotective panel of olive polyphenols: anti-demyelination and anti-stroke properties and mechanisms of action".

Reviewer 4   -   Comments to Authors

Grubić Kezele et.al did a great job introducing different types of Polyphenols and their effects on anti-demyelination and anti-stroke properties. The review paper is well structured and well written and ready for publish.

Response to the Reviewer:

We thank the reviewer for the very nice comments. 

We thank the editors and reviewers for their observations, suggestions, and valuable comments. We greatly appreciate it and sincerely hope that the editors and reviewers will be pleased with the revision of our manuscript.

Thank you for your consideration, we are looking forward to your answer.

Sincerely,

the authors.

Round 2

Reviewer 1 Report

The authors' responses/revisions to my original evaluation are fair. In an ideal context, I would suggest more informative illustrations and tables to best guide the reader. There are some remaining errors with eNOS being referred to as "inducible" on line 388 (that's actually iNOS) and "regulatation" in Figure 2 (should be "regulation"). If images in Figures 1 & 2 are not original (branches, brain, or olive oil container) l, I would secure copyright permissions as well.

Author Response

Reviewer 1   -   Comments to Authors:

The authors' responses/revisions to my original evaluation are fair. In an ideal context, I would suggest more informative illustrations and tables to best guide the reader.

Response to the Reviewer:

We thank the reviewer for his/her comments. Yes, we made the graphical abstract (GA) that will help attract more readers and increase the visibility of your paper once it is published online. In addition to summarizing the content, it should represent the topic of the article in an attentiongrabbing way. The GA is totally made from our ideas. We are the authors of all GA parts.

There are some remaining errors with eNOS being referred to as "inducible" on line 388 (that's actually iNOS) and "regulatation" in Figure 2 (should be "regulation").

Response to the Reviewer:

We thank the reviewer for pointing out these errors. The figure has been corrected, i.e., "regulatation" to "regulation" and "resistence" to "resistance." We have also rechecked all words in both figures, including GA. We deleted „inducible“ as it is „endothelial“: Line 388.

Reviewer 3 Report

I would like to thank the authors for taking into account the comments. The manuscript is suitable for publication in its current form.

Author Response

Reviewer 3   -   Comments to Authors

I would like to thank the authors for taking into account the comments. The manuscript is suitable for publication in its current form.

Response to the Reviewer:

We thank the reviewer very much.
